# Remote monitoring of COVID-19 positive high-risk patients in domestic isolation: A feasibility study

David Wurzer[1☉], Paul Spielhagen[1☉], Adonia Siegmann[1], Ayca Gercekcioglu[1], Judith Gorgass[1], Simone Henze[1], Yuron Kolar[1], Felix Koneberg[2], Sari Kukkonen[1], Hannah McGowan[1], Stefanie Schmid-Eisinger[1], Alexander Steger[3], Michael Dommasch[3], Hans Ulrich Haase[3], Alexander Müller[3], Eimo Martens[3], Bernhard Haller[4], Katharina M. Huster[3], Georg Schmidt[3,5]*

1 Telecovid Centre, Klinikum Rechts der Isar, Technical University of Munich, Munich, Germany, 2 Cosinuss GmbH, Munich, Germany, 3 Department of Internal Medicine I, Klinikum Rechts der Isar, Technical University of Munich, Munich, Germany, 4 Institute of Medical Informatics, Statistics and Epidemiology, Technical University of Munich, Munich, Germany, 5 DZHK (German Centre for Cardiovascular Research), Partner Site Munich Heart Alliance, Munich, Germany

☉ These authors contributed equally to this work.
* gschmidt@tum.de

**Data Availability Statement:** We uploaded our Minimal Data Set, which is the basis for the data shown in the paper, at https://mediatum.ub.tum.

## Abstract

### Background

If a COVID-19 patient develops a so-called severe course, he or she must be taken to hospital as soon as possible. This proves difficult in domestic isolation, as patients are not continuously monitored. The aim of our study was to establish a telemonitoring system in this setting.

### Methods

Oxygen saturation, respiratory rate, heart rate and temperature were measured every 15 minutes using an in-ear device. The data was transmitted to the Telecovid Centre via mobile network or internet and monitored 24/7 by a trained team. The data were supplemented by daily telephone calls. The patients´ individual risk was assessed using a modified National Early Warning Score. In case of a deterioration, a physician initiated the appropriate measures. Covid-19 Patients were included if they were older than 60 years or fulfilled at least one of the following conditions: pre-existing disease (cardiovascular, pulmonary, immunologic), obesity (BMI >35), diabetes mellitus, hypertension, active malignancy, or pregnancy.

### Findings

153 patients (median age 59 years, 77 female) were included. Patients were monitored for 9 days (median, IQR 6–13 days) with a daily monitoring time of 13.3 hours (median, IQR 9.4–17.0 hours). 20 patients were referred to the clinic by the Telecovid team. 3 of these required intensive care without invasive ventilation, 4 with invasive ventilation, 1 of the latter died. All patients agreed that the device was easy to use. About 90% of hospitalised patients indicated that they would have delayed hospitalisation further if they had not been part of the study.

de/1622036. For further data, anyone interested may contact me or the Ethics Committee (addresses found on the URL).

**Funding:** This work was supported by the TUM-Universitätsstiftung, by the Margarete-Ammon-Stiftung and by the Bayerisches Staatsministerium für Wissenschaft und Kunst (U.7-H4001.1.7/35/3). The sponsors had no role in study design, data collection, data analysis, data interpretation, or writing of the report. Cosinuss GmbH, Munich, Germany provided support in the form of salaries for FK, but did not have any additional role in the study design, data collection and analysis, decision to publish, or preparation of the manuscript. The specific role of this author is articulated in the 'author contributions' section.

**Competing interests:** FK is an employee of cosinuss, the manufacturer of the ear sensor used in this study. This affiliation does not alter our adherence to PLOS ONE policies on sharing data and materials. All other authors declare no competing interests.

## Interpretation

Our study demonstrates the successful implementation of a remote monitoring system in a pandemic situation. All clinically necessary information was obtained and adequate measures were derived from it without delay.

## Introduction

Infections with severe acute respiratory syndrome coronavirus type 2 (SARS-CoV-2) threaten to overburden public health systems in many countries [1–3].

In developed countries, a vast majority of SARS-CoV-2-positive patients are placed in home isolation immediately after diagnosis. In Germany, e.g., this rate is around 90% [4]. A considerable number of these patients in domestic isolation will develop a severe course of the disease, especially patients with risk factors such as advanced age, cardiovascular, pulmonary, or metabolic diseases.

Effective self-monitoring in domestic isolation is rather unfeasible. Some health authorities recommend that affected patients take their temperature two to three times a day and document their self-perceived symptoms regularly. Recording of objective parameters such as SpO2, heart rate or the respiratory rate, which are proven to lower mortality if monitored early, is not performed. The lack of objective vital parameters can result in delayed emergency hospital admission of patients with progressive disease [5].

If admission is delayed because the transition into a severe course is not recognised in time, the non-invasive supportive measures may not be sufficient and the patients will require intensive care [6]. The stay in the intensive care unit (ICU) can last for weeks and the chances for survival is significantly reduced. Especially, if a patient needs invasive ventilation, the prognosis is considered poor [7]. Thus, it is therefore highly desirable to avoid such costly, often unsuccessful course.

To relieve health care systems and to improve the overall prognosis of COVID-19 patients, it is urgently needed to identify patients at an early stage of disease progression so that they can be admitted to hospital in a timely manner.

A continuous monitoring of vital parameters in high-risk patients in domestic isolation would pave the way to objectively assess the condition of the patient. An affordable monitoring system that is easy to use at home could fill a crucial gap in managing COVID-19 patients.

In this paper, we present such a remote monitoring system that is low cost and practical, by using resources that are readily available. The aim of this paper is to demonstrate the technical and logistical feasibility of this Telecovid system.

## Methods

### Study population

Between April 2020 and April 2021, patients with a positive SARS-CoV-2 PCR test were included if they were older than 60 years of age or if they fulfilled at least one of the following conditions: pre-existing disease (cardiovascular, pulmonary, immunologic), obesity (BMI >35), diabetes mellitus, hypertension, active malignancy, or pregnancy. All patients gave written informed consent. The patients were instructed to wear the in-ear device continuously during home isolation. Remote monitoring was stopped with the end of isolation or with hospitalisation. At that time point, the participants were contacted once more by

phone. They were asked about their subjective perception of the illness and opinion about their participation in the study by means of a questionnaire. The ethics committees of the Ludwig-Maximilians University and of the Technical University Munich approved the study protocol. The study was conducted in accordance with the principles of the Declaration of Helsinki.

## In-ear monitoring device

A plethora of different non-invasive devices for monitoring biosignals are available, including wearables of all kinds (smartwatches, smartphones, etc.) and so-called skin-attachable electronics. Many of these devices are based on photoplethysmographic signals (PPG).

The aim of our study was to reliably and continuously monitor biosignals relevant to the course of an infectious disease. Therefore, we sought a device that allows for robust recording of the PPG signal. We decided in favour of the in-ear device cosinuss ˚One, manufactured by Cosinuss GmbH, Munich, Germany. The necessary in ear devices were purchased; the company had no further involvement in our study. The advantage of the in-ear device is its position in the external ear canal which provides protection from disturbing stray light and a relatively stable position for the sensor (Fig 1). Besides that, the body temperature can be reliably taken in the external ear canal. The handling of the device is simple, similar to that of a hearing aid and is therefore also well suited for older, technically inexperienced patients.

The monitoring system consists of the in-ear wearable device and a data gateway that receives data from the device via Bluetooth, stores it and transmits it to a server via the mobile network or internet.

The in-ear monitoring device is equipped with two sensors: a photo diode for reading the PPG with red and infrared LEDs and a contact thermometer for reading the temperature.

The sampling frequency of the PPG signal is 200 Hz, the sampling depth 19 bits. The core body temperature is recorded with a sampling frequency of 1 Hz and an accuracy of 0.3˚C. SpO2, heart rate and the respiratory rate are extracted from the PPG.

## Data transfer and data protection

After the patient has inserted the in-ear device and plugged in the gateway, the sensor automatically starts measuring biosignals every fifteen minutes for three minutes. Additional measurements can be manually triggered by the Telecovid Centre at any time. The data is transmitted to the gateway via Bluetooth 4.2 Low Energy and from there sent in encrypted form via the mobile phone network or the internet to a secure server.

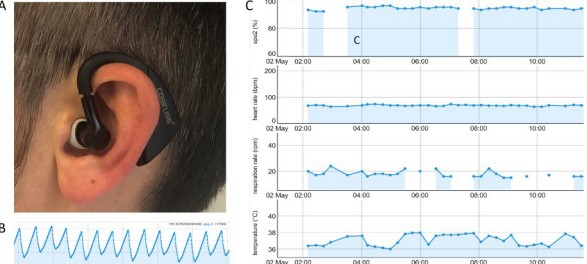

**Fig 1. In-ear monitoring device and measurements.** A: Device placed in the external ear canal; B: undisturbed photoplethysmogram (PPG), used for quality control; C: Section of the course of a patient's vital signs (SpO2, heart rate, respiratory rate and core temperature) over the course of several hours.

The in-ear device needs charging for one hour per 24-hour period. The distance between the in-ear device and the gateway should not exceed ten meters in order to maintain the Bluetooth connection between the two devices.

Personal data is pseudonymised and data transmission is encrypted via Transport Layer Security (TLS). The Dash/Web interface ensures that only authorised persons have access to the data. Data access is restricted to authorized members of the Telecovid team with a dedicated password-protected user account. The data is stored on an internal server within the hospital data information system of the Klinikum rechts der Isar of the Technical University of Munich.

## PPG and transmission issues

Due to the close proximity of the external ear canal and the temporomandibular joint, data quality often deteriorates while the patient is eating or talking. In this case, the next measurement is to be waited for, before further action is taken. Data quality may also be poor if the optical sensor is stained from ear wax. In this case, the unit's photo sensor must be cleaned with a dry cotton cloth.

In addition, there may be several technical issues that can negatively affect the data quality. For instance, if the device does not fit the diameter of the ear canal properly, it may lead to inaccurate data readings. If the device is too large, it cannot be placed deep enough inside the ear and if it is too small, it may wobble back and forth. In these instances, the device must be replaced by a device of an appropriate size. There are three device sizes to ensure a proper fit. The appropriate size is determined with disposable sensor tip templates.

Very rarely, there may be mechanical defects affecting the device. In this case, the PPG signal turns into white noise with a very low amplitude resulting in a complete loss of usable signal.

Issues can also occur in the transmission of data, both in the Bluetooth connection between the device and the gateway and in the transmission between the gateway and the server via mobile phone network. The former issue happens when the patient moves too far away from the gateway or if the connectivity is disturbed by walls or other massive obstacles. The latter occurs when the mobile network connection is of poor quality. If LAN or WLAN is available in the patient's home, the gateway can be connected to it instead to ensure a timely data transmission.

If these measures do not lead to a satisfactory result, a team member will drive to the patient, use a reference pulse oximeter to measure the oxygen saturation and check all corrective measures again on the spot.

## Individual risk assessment

After each single measurement, the patient's individual risk is assessed on the basis of a modified National Early Warning Score automatically and presented on the dashboard [8]. Our modified score takes into account SpO2, respiratory rate, heart rate and temperature (see Table 1). A score of 3 or below indicates low risk, a score of 4 to 5 indicates intermediate risk and a score of 6 or higher indicates high risk.

**Table 1. Modified national early warning score [8].**

|  | 3 | 2 | 1 | 0 | 1 | 2 | 3 |
|---|---|---|---|---|---|---|---|
| SpO2 (%) | ≤ 91 | 92–93 | 94–95 | ≥ 95 |  |  |  |
| Respiratory rate (min⁻¹) | ≤ 8 |  | 9–11 | 12–20 |  | 21–24 | ≥ 25 |
| Heart rate (min⁻¹) | ≤ 40 |  | 41–50 | 51–90 | 91–110 | 111–130 | ≥ 130 |
| Temperature (˚C) | ≤ 35.0 |  | 35.1–36.0 | 36.1–38.0 | 38.1–39.0 | ≥ 39.0 |  |

In order to be able to provide 24/7 monitoring, the team consisted of 25 to 35 trained members, with a maximum of four present at the COVID centre at any given time. Each staff member can monitor up to 20 patients at a time. A team of four physicians supports the Telecovid team and consults with them once a day and checks the data. In addition, the physician on duty is available around the clock to answer questions in the event of a patient's deterioration, i.e. a National Early Warning Score of $\geq 6$, and to then initiate the appropriate measure. At least once a day, the study participants are contacted via the phone. During these regular calls, a structured interview containing a questionnaire of COVID-19 symptoms is used. The questionnaire includes symptoms such as cough, cold, loss of smell and taste, headache, pain in the limbs and shortness of breath. In particular, the team also checks whether the study participants show cognitive abnormalities or appear disoriented.

If there is suspicion that a patient's health is deteriorating, (risk score > = 6) or in case of inconclusive findings (risk score 4–5), the patient is also called outside the regular calls. A decision for immediate hospital admission is always made after consultation with the patient and / or his relatives and is initiated by the responsible physician of the team.

## Results

The Telecovid study included a total of 153 patients (77 female) residing in Bavaria, with a focus on the Munich area. Median age was 59 years (interquartile range (IQR) 42–68 years). 66 patients were "older than 60 years", the oldest participant was 94 years old. The frequency of underlying conditions is shown in Table 2. The time between the onset of symptoms and inclusion in the study was 3 days (median, IQR 1–4).

On average, patients were monitored for 9 days (median, IQR 6–13 days). The daily monitoring time was 13.3 hours (median, IQR 9.4–17.0). The PPG raw signals were of satisfactory quality in two thirds of the measurements. In the other instances, the PPG signal was temporarily non-analyzable due to disruptions induced by chewing or speaking or when the Bluetooth transmission was temporarily halted. These disruptions usually occurred towards the end of the isolation, when the patients' condition had improved enough to allow them to talk, eat and move freely in the home. In both instances, a re-measurement is manually triggered, if necessary after contacting the patient by phone to get valid results. In the event of a data disruption of more than one hour, the patient was always contacted to make sure that his condition had not worsened. During sleep (midnight to 6 am), about 83% of the measurements were of satisfactory quality, during the day about 55%.

In rare cases, continuous transmission of the data was not possible due to poor mobile network availability. Particularly at locations supplied only with a 2G network, considerable

**Table 2. Frequency of risk factors at inclusion.**

| Risk factor | Frequency (n) | Frequency (%) |
| --- | --- | --- |
| Age >60 yrs | 66 | 43.1 |
| Female sex | 77 | 50.3 |
| Cardiovascular disease | 57 | 37.3 |
| Pulmonary disease | 32 | 20.9 |
| Body mass index >35 | 28 | 18.3 |
| Diabetes mellitus | 21 | 13.7 |
| Immunologic disease | 20 | 13.1 |
| Depression | 20 | 13.1 |
| Active malignancy | 16 | 10.5 |
| Pregnancy | 3 | 2.0 |

delays in data transmission and sometimes even a complete failure of transmission were observed. If a LAN or WLAN was available in the patient's home, the problem was solved by connecting the gateway to the server via internet.

## Hospital admissions and outcome

A total of 20 patients (13%) were referred to hospital by the Telecovid team. The interval between the positive SARS-CoV-2 PCR test and hospital admission was 6 days on average (median, IQR 3–11). Patients were hospitalised for a median of 10 days (IQR 4–19 days, maximum 41 days). 13 out of the 20 patients were managed at the normal ward with supportive measures alone. They remained there for a median of 6 days (IQR 4–9).

Seven patients required intensive medical treatment, three of them were temporarily on invasive ventilation. A fourth patient died after 25 days of invasive ventilation. This patient had been enrolled in the study only 24 hours before the transfer to the clinic. Initially his conditions appeared non-critical, but after 24 hours, we initiated hospitalisation because his SpO2 values dropped below 80%. Two days after hospitalisation he was transferred to the intensive care unit and invasive ventilation was required.

None of the 133 patients monitored throughout their domestic isolation suffered an unforeseen complication.

Patients admitted to the clinic were on average older (mean 63.4 years versus 55.2 years) and more likely to suffer from underlying diseases (cardiovascular 1.6-fold, respiratory 1.4-fold, active malignancy or immunologic disease 1.8-fold, obesity 3-fold increased). In terms of symptoms, they showed fever and dyspnea more frequently (2.3-fold and 1.4-fold increase, respectively). The last measured NEWS score in home isolation averaged 5.4 ±1.3 (mean, standard deviation). In a group matched by gender and age of non-admitted patients, these figures were 1.9 ±2.2. The average of the last SpO2 values before hospitalisation was 88.0% ±6.7%. This was in stark contrast to measurements of an age and sex matched group of not hospitalised patients who showed an SpO2 average of 96.5% ±4.0%.

At the end of domestic isolation, patients were interviewed by means of a questionnaire on their experience. All agreed that the device was easy to use (68% 'strongly agree' and 32% 'somewhat agree'). The statements about the comfort level of wearing the device were more ambiguous: while two out of three participants answered positively (23%) or somewhat positively (45%), 18% were 'not sure' and 14% 'somewhat disagreed'.

However, the vast majority of patients stated that they felt positive about participating in the study. The question "Did the study participation improve your subjective well-being?" was answered positively by the majority (82% "strongly agree"). The agreement was even greater for the question "Did you feel safe during the study? (91% "strongly agree"). Similarly, 91% of the participants stated that they would highly recommend the study participation to other patients.

About 90% of hospitalised patients indicated that they would have delayed hospitalisation further if they had not been part of the study, even the majority of patients who were transferred to the ICU had not realized the severity of their condition. After discharge from hospital, all patients declared that they were convinced that the hospitalisation prompted by the Telecovid team had improved their chances of recovery.

## Discussion

To the authors' knowledge, this is the first study to monitor patients continuously and remotely in domestic isolation and, in the event of a critical health deterioration, arrange for immediate hospital admission. The novelty of the study stems from its complete telemedicine

format, i.e. all observation, monitoring, and counselling up to the point of possible admission, was handled remotely, whereas patients remained at home in domestic isolation.

Our findings prove that with current technology, a remote observation network for COVID-19 patients is feasible. In our study, all necessary data could be collected with sufficient quality and safely transferred to the Telecovid Centre for evaluation.

Additionally, our results show that meaningful medical care is possible using this remote system. In this study, vital parameters of patients in home isolation were continuously monitored, evaluated and interpreted by the team, the patients were interviewed about their health status at least once a day over the phone and counselled according to the findings. In this way, all clinically necessary information could be obtained, adequate measures could be derived from it without the need for face-to-face contact. None of the patients we monitored in home isolation came into an immediate emergency situation.

All patients admitted by the Telecovid team were hospitalised after an initial examination in the emergency room, indicating the admission was justified. None of the patients required ventilation on admission, thus, their clinical deterioration was recognised by us in time.

The patients were very satisfied with the telemedical care and felt safer because of it. A minority of the participants criticised the poor comfort of the device. Our rationale for choosing the in-ear device was because it was less susceptible to interference and, unlike other devices, it allows temperature measurement.

Since only Bluetooth, mobile net or internet are required, the data transfer from the patient to the Telecovid Centre does not require any costly investments in new technology as these are available in most parts of the world including developing countries. However, in rural areas with mobile network coverage below the 3G standard, there were repeatedly significant delays in data transmission or even transmission failures, so that continuous data collection was not always possible. One solution would be to adapt the device used for this study in a way that the raw data is not transmitted in full resolution. In the long term, of course, the nationwide expansion of the mobile network to at least 3G is desirable.

We assume that in the event of a pandemic, such remote monitoring will help to relieve the burden on general practitioners and medical officers. Patients can be well cared for by telemedicine without the need for direct contact with a doctor or a nurse. This means that no unnecessary resources are bound and, additionally, the risk of infection for medical and nursing staff can be reduced. At the same time, the patient's chances of survival are likely to be increased, as their clinical course is continuously monitored. Moreover, the methodology is transferable to other health issues as fundamentally clinically important vital signs (SpO2, respiratory rate, heart rate and temperature) are recorded.

## Limitations

It must be emphasised that this is a feasibility study. Statements on the effectiveness in terms of preventing severe courses of disease or reducing deaths are not possible. However, we do assume the patient's chances of survival are likely to be increased, as a deterioration might be detected earlier and thus the length of hospital stay and the mortality might be reduced. Proving such superiority requires a large-scale comparative study between telemedical monitoring and conventional care.

We also assume that the remote monitoring of patients will directly save resources in the doctors´ offices and at the health department and, in the course of time, also preserve resources through shorter hospital stays and the avoidance of invasive ventilation. Again, no data are available concerning this hypothesis and it would need to be tested in a future study with a much larger sample size.

## Conclusion

We are convinced that we have made an important step towards future clinical care options with our Telecovid study. We were able to show that it is possible to obtain all clinically necessary information via a remote monitoring system and to derive adequate measures from it without delay. The data transfer technology required is available practically everywhere; the system is therefore suitable for being scaled up promptly if required.

## Author Contributions

**Conceptualization:** Georg Schmidt.

**Data curation:** David Wurzer, Paul Spielhagen, Adonia Siegmann, Ayca Gercekcioglu, Judith Gorgass, Simone Henze, Yuron Kolar, Felix Koneberg, Sari Kukkonen, Hannah McGowan, Stefanie Schmid-Eisinger, Alexander Müller.

**Formal analysis:** Felix Koneberg, Hannah McGowan, Stefanie Schmid-Eisinger, Alexander Steger, Alexander Müller, Bernhard Haller, Georg Schmidt.

**Funding acquisition:** Simone Henze, Georg Schmidt.

**Investigation:** David Wurzer, Paul Spielhagen, Adonia Siegmann, Ayca Gercekcioglu, Judith Gorgass, Simone Henze, Yuron Kolar, Sari Kukkonen, Hannah McGowan, Stefanie Schmid-Eisinger, Alexander Steger, Michael Dommasch, Hans Ulrich Haase, Eimo Martens, Georg Schmidt.

**Methodology:** Sari Kukkonen, Alexander Steger, Michael Dommasch, Hans Ulrich Haase, Eimo Martens, Georg Schmidt.

**Project administration:** David Wurzer, Paul Spielhagen, Adonia Siegmann, Ayca Gercekcioglu, Judith Gorgass, Simone Henze, Yuron Kolar, Sari Kukkonen, Hannah McGowan, Stefanie Schmid-Eisinger, Alexander Steger, Eimo Martens, Katharina M. Huster, Georg Schmidt.

**Resources:** David Wurzer, Paul Spielhagen, Adonia Siegmann, Ayca Gercekcioglu, Judith Gorgass, Simone Henze, Yuron Kolar, Sari Kukkonen, Hannah McGowan, Hans Ulrich Haase, Alexander Müller, Georg Schmidt.

**Software:** Felix Koneberg, Alexander Müller, Bernhard Haller.

**Supervision:** Michael Dommasch, Georg Schmidt.

**Validation:** Alexander Steger, Georg Schmidt.

**Visualization:** Felix Koneberg, Alexander Müller.

**Writing – original draft:** David Wurzer, Paul Spielhagen, Georg Schmidt.

**Writing – review & editing:** David Wurzer, Paul Spielhagen, Adonia Siegmann, Hannah McGowan, Bernhard Haller, Katharina M. Huster, Georg Schmidt.

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
