## [Decision Letter · Decision Letter 0]

18 Jun 2021

PONE-D-21-17053

Remote monitoring of COVID-19 positive high-risk patients in domestic isolation: A feasibility study

PLOS ONE

Dear Dr. Schmidt,

Thank you for submitting your manuscript to PLOS ONE. After careful consideration, we feel that it has merit but does not fully meet PLOS ONE’s publication criteria as it currently stands. Therefore, we invite you to submit a revised version of the manuscript that addresses the points raised during the review process.

Please address the issues and revise accordingly.

We look forward to receiving your revised manuscript.

Kind regards,

Academic Editor

PLOS ONE

Journal Requirements:

We note that one or more of the authors are employed by a commercial company: cosinuss° GmbH.

2.1. Please provide an amended Funding Statement declaring this commercial affiliation, as well as a statement regarding the Role of Funders in your study. If the funding organization did not play a role in the study design, data collection and analysis, decision to publish, or preparation of the manuscript and only provided financial support in the form of authors' salaries and/or research materials, please review your statements relating to the author contributions, and ensure you have specifically and accurately indicated the role(s) that these authors had in your study. You can update author roles in the Author Contributions section of the online submission form.

2.2. Please also provide an updated Competing Interests Statement declaring this commercial affiliation along with any other relevant declarations relating to employment, consultancy, patents, products in development, or marketed products, etc.  

Reviewers' comments:

Reviewer's Responses to Questions

**Comments to the Author**

1. Is the manuscript technically sound, and do the data support the conclusions?

Reviewer #1: Yes

Reviewer #2: Yes

2. Has the statistical analysis been performed appropriately and rigorously? 

Reviewer #1: Yes

Reviewer #2: N/A

3. Have the authors made all data underlying the findings in their manuscript fully available?

Reviewer #1: Yes

Reviewer #2: Yes

4. Is the manuscript presented in an intelligible fashion and written in standard English?

Reviewer #1: Yes

Reviewer #2: Yes

5. Review Comments to the Author

Reviewer #1: This article is entitled “Remote monitoring of COVID-19 positive high-risk patients in domestic isolation: A feasibility study”. The authors had committed to a great task, to monitor the COVID-19 positive patients and transport to hospital early if deterioration detected. Some questions were raised and listed below:

1. Abstract: nil

2. Introduction:nil

3. Methods:

甲、 The authors may describe the works of TeleCOVID team. How many staffs were there? Did the National Early Warning Score (NEWS) calculate automatically? Or the team members should calculate the score? How many times did the team members call outside regular calls?

乙、 The stopping criteria were not mentioned in the manuscript.

4. Results:

甲、 Please create a table describing the demographics of the enrolled subjects, including age, gender, underlying diseases, symptoms, and the reasons for end of monitoring.

乙、 Please describe the mean and range of NEWS in non-hospitalized and hospitalized subjects, respectively.

丙、 In terms of NEWS, did the authors suggest SpO2 particularly important? How many subjects who were sent to hospitals were determined by mainly SpO2?

丁、 Could the authors calculate the time between detection of alarming score (4-5, or ≥6) and the initiation of transportation to hospital. Since there were remarkable amount of data disruption, especially during the day, the readers would be worry about the impact on delaying transference.

5. Discussions

甲、 Since there were no subjects older than 69 years, the authors may add a limitation that the feasibility study was not done in patients ≥69 years old.

Reviewer #2: The authors presented a remote monitoring system utilizing an in-ear device connected via Bluetooth signals to the gateway for transmission to the Telecovid center using either mobile network or WIFI. This monitoring is coupled with daily calls and with the help of a team of trained personnel and physicians.

This reviewer accesses this manuscript from the angle and scope of the above situation and the limitations as described by the authors.

IN general, this is a well written manuscript.

Comment 1: The conditions, scores, age, pre-existing conditions or diseases for the patients referred to the hospitals should be tabulated and analyzed, including those who were admitted into ICU, with and without invasive ventilation.

Comment 2: The author are using a modified NEWS based on reference 8. It would be useful to state the basis of the modification in relation to COVID-19 symptoms and risk factors.

6. PLOS authors have the option to publish the peer review history of their article (what does this mean?). If published, this will include your full peer review and any attached files.

Reviewer #1: **Yes: **Shu-Hsing Cheng

Reviewer #2: No

---

## [Author Response · Author response to Decision Letter 0]

31 Jul 2021

45. Review Comments to the Author

Reviewer #1: This article is entitled “Remote monitoring of COVID-19 positive high-risk patients in domestic isolation: A feasibility study”. The authors had committed to a great task, to monitor the COVID-19 positive patients and transport to hospital early if deterioration detected. Some questions were raised and listed below:

1. Abstract: nil

2. Introduction:nil

3. Methods:

甲、 The authors may describe the works of TeleCOVID team. How many staffs were there? 

Thank you for this helpful question. In order to be able to estimate the resources required for such a task, it is necessary to know the personnel requirements: To be able to provide 24/7 monitoring, the team consisted of 25 to 35 trained members, with a maximum of 4 present at the COVID centre at any given time. Four physicians supported the team. The physician on duty performed the telephone rounds and was on call. We included this information in the manuscript (METHODS, individual risk assessment).

Did the National Early Warning Score (NEWS) calculate automatically? Or the team members should calculate the score? How many times did the team members call outside regular calls?

The modified National Early Warning Score is calculated automatically and shown on the dashboard. We included this important information in the manuscript (METHODS, individual risk assessment). Phone calls were performed up to four times in 24 hours, depending on the health status of the patient. 

乙、 The stopping criteria were not mentioned in the manuscript.

Thank you for this important comment. Remote monitoring was stopped with the end of isolation, unless the participant wished to continue monitoring for a few more days. This was the case for five participants. We have added the information on the ending of the monitoring period in the manuscript (METHODS, study population).

4. Results:

甲、 Please create a table describing the demographics of the enrolled subjects, including age, gender, underlying diseases, symptoms, and the reasons for end of monitoring.

We created a table to further describe the patients´ characteristics and included it in the manuscript (RESULTS, Table 2) and described them in the RESULTS section.

乙、 Please describe the mean and range of NEWS in non-hospitalized and hospitalized subjects, respectively.

In patients admitted to the hospital, the last measured NEWS score in home isolation averaged 5.4 ±1.3 (mean, standard deviation). In a matched group of non-admitted patients by gender and age, these figures were 1.9 ±2.2. This is, together with the corresponding SpO2 values, now mentioned in the text.

丙、 In terms of NEWS, did the authors suggest SpO2 particularly important? How many subjects who were sent to hospitals were determined by mainly SpO2?

This is a very interesting question. Indeed, SpO2 was the most important parameter. Many patients did not yet feel severely impaired, but already had a reduced SpO2 (silent hypoxia). The decision to hospitalize was always a medical decision based on all available findings. However, SpO2 was the key parameter and no patient who was admitted had an SpO2 above 91%. We analysed the last three measurements of SpO2 for the hospitalised patients and compared these numbers to the measurements of an age and sex matched group of not hospitalised patients. The hospitalised had an SpO2 of 88.0% ±6.7%, the control group of 96.5% ±4.0%.

丁、 Could the authors calculate the time between detection of alarming score (4-5, or ≥6) and the initiation of transportation to hospital. Since there were remarkable amount of data disruption, especially during the day, the readers would be worry about the impact on delaying transference.

We clearly see the point made here. We did not record how long the period was between deterioration and inpatient admission. In case of suspicion of a possible worsening, we always contacted the patient immediately, latest within 60 minutes. We are confident that we did not miss any critical deterioration of the patient's condition. We included this information in the manuscript (RESULTS).

5. Discussions

甲、 Since there were no subjects older than 69 years, the authors may add a limitation that the feasibility study was not done in patients ≥69 years old.

We assume that this is a misunderstanding. In the manuscript, we report the interquartile range, not the entire age distribution. 

There were 36 participants >= 69 years old, the oldest participant was 94 years old. 66 participants were “older than 60 years”, six were 60 years old. To further clarify this issue, we explained the data in more detail in the RESULTS and also added a table of patients´ characteristics (Table 2).

Reviewer #2: The authors presented a remote monitoring system utilizing an in-ear device connected via Bluetooth signals to the gateway for transmission to the Telecovid center using either mobile network or WIFI. This monitoring is coupled with daily calls and with the help of a team of trained personnel and physicians.

This reviewer accesses this manuscript from the angle and scope of the above situation and the limitations as described by the authors.

In general, this is a well written manuscript.

Comment 1: The conditions, scores, age, pre-existing conditions or diseases for the patients referred to the hospitals should be tabulated and analyzed, including those who were admitted into ICU, with and without invasive ventilation.

We further analysed the patients´ characteristics and described them briefly in the manuscript (RESULTS, Hospital admission and outcome, and Table 2). Additionally, we analysed the average of the last three measurements of SpO2 before admission to hospital in comparison to an age and sex matched group of patients not admitted to the clinic. The hospitalised group had an SpO2 of 88.0% ±6.7% on average, while the control group had 96.5% ±4.0%. This is also integrated in the RESULT section. The differences we observed between the groups are well in line with previously published data. We would like to refrain from a more detailed presentation because we are concerned that this may distract from our main message, which is that remote monitoring is feasible.

Comment 2: The authors are using a modified NEWS based on reference 8. It would be useful to state the basis of the modification in relation to COVID-19 symptoms and risk factors.

Thank you for your comment. We had to exclude the parameter “blood pressure” as the measurement was not possible in the remote monitoring and the AVPU score was only indirectly measured by the daily phone call. It was also not taken in account for the calculation.

---

## [Decision Letter · Decision Letter 1]

24 Aug 2021

Remote monitoring of COVID-19 positive high-risk patients in domestic isolation: a feasibility study

PONE-D-21-17053R1

Dear Dr. Schmidt,

We’re pleased to inform you that your manuscript has been judged scientifically suitable for publication and will be formally accepted for publication once it meets all outstanding technical requirements.

Kind regards,

Academic Editor

PLOS ONE

Additional Editor Comments (optional):

Reviewers' comments:

Reviewer's Responses to Questions

**Comments to the Author**

1. If the authors have adequately addressed your comments raised in a previous round of review and you feel that this manuscript is now acceptable for publication, you may indicate that here to bypass the “Comments to the Author” section, enter your conflict of interest statement in the “Confidential to Editor” section, and submit your "Accept" recommendation.

Reviewer #1: All comments have been addressed

Reviewer #2: All comments have been addressed

2. Is the manuscript technically sound, and do the data support the conclusions?

Reviewer #1: Yes

Reviewer #2: Yes

3. Has the statistical analysis been performed appropriately and rigorously? 

Reviewer #1: Yes

Reviewer #2: Yes

4. Have the authors made all data underlying the findings in their manuscript fully available?

Reviewer #1: Yes

Reviewer #2: Yes

5. Is the manuscript presented in an intelligible fashion and written in standard English?

Reviewer #1: Yes

Reviewer #2: Yes

6. Review Comments to the Author

Reviewer #1: Congratulations to the authors. The authors' responses were very comprehensive, and every query had been answered. I did not have further comments.

Reviewer #2: (No Response)

7. PLOS authors have the option to publish the peer review history of their article (what does this mean?). If published, this will include your full peer review and any attached files.

Reviewer #1: **Yes: **Shu-Hsing Cheng

Reviewer #2: No

---

## [Editor Report · Acceptance letter]

17 Sep 2021

PONE-D-21-17053R1 

Remote monitoring of COVID-19 positive high-risk patients in domestic isolation: a feasibility study 

Dear Dr. Schmidt:

I'm pleased to inform you that your manuscript has been deemed suitable for publication in PLOS ONE. Congratulations! Your manuscript is now with our production department. 

Kind regards, 

on behalf of

Dr. Robert Jeenchen Chen 

Academic Editor

PLOS ONE